# Doxycycline for the prevention of progression of COVID-19 to severe disease requiring intensive care unit (ICU) admission: A randomized, controlled, open-label, parallel group trial (DOXPREVENT.ICU)

**Raja Dhar[1‡], John Kirkpatrick[2‡], Laura Gilbert[3], Arjun Khanna[4], Mahavir Madhavdas Modi[5], Rakesh K. Chawla[6], Sonia Dalal[7], Venkata Nagarjuna Maturu[8], Marcel Stern[9], Oliver T. Keppler[9], Ratko Djukanovic[10,11‡], Stephan D. Gadola[10,11,12‡]***

1 Department of Pulmonology, CMRI Hospital, Kolkata, India, 2 John Kirkpatrick, MSc, Independent Researcher, Cambridgeshire, United Kingdom, 3 Laura Gilbert, Rutherford Research, Hampshire, United Kingdom, 4 Pulmonary and Critical care medicine, Yashoda Superspeciality Hospital, Kaushambi, Ghaziabad, UP, India, 5 Ruby Hall Clinic, Pune, Maharashtra, India, 6 Saroj Super Speciality Hospital and Jaipur Golden Hospital, Dept of Respiratory Medicine, Critical Care and Sleep Disorders, New Delhi, India, 7 Sterling Hospital and Kalyan Hospital, Vadodara, India, 8 Yashoda Hospitals, Hyderabad, India, 9 Max von Pettenkofer Institute and Gene Center, Virology, National Reference Center for Retroviruses, Faculty of Medicine, LMU München, Munich, Germany, 10 Clinical and Experimental Sciences, Faculty of Medicine, University of Southampton, Southampton, United Kingdom, 11 NIHR Southampton Biomedical Research Centre, Southampton, United Kingdom, 12 Rheumatology and Pain Medicine, Bethesda Hospital, Basel, Switzerland

‡ RD and JK share first authorship on this work. RD and SDG are joint senior authors on this work.
* Stephan.Gadola@bethesda-spital.ch

## Abstract

### Background

After admission to hospital, COVID-19 progresses in a substantial proportion of patients to critical disease that requires intensive care unit (ICU) admission.

### Methods

In a pragmatic, non-blinded trial, 387 patients aged 40–90 years were randomised to receive treatment with SoC plus doxycycline (n = 192) or SoC only (n = 195). The primary outcome was the need for ICU admission as judged by the attending physicians. Three types of analyses were carried out for the primary outcome: "Intention to treat" (ITT) based on randomisation; "Per protocol" (PP), excluding patients not treated according to randomisation; and "As treated" (AT), based on actual treatment received. The trial was undertaken in six hospitals in India with high-quality ICU facilities. An online application serving as the electronic case report form was developed to enable screening, randomisation and collection of outcomes data.

### Results

Adherence to treatment per protocol was 95.1%. Among all 387 participants, 77 (19.9%) developed critical disease needing ICU admission. In all three primary outcome analyses,

**Data Availability Statement:** All data underlying the results presented in the study are available

online on GitHub (https://github.com/PuzzledFace/DoxyICU_Results).

**Funding:** RDh received an unrestricted grant (grant/study code: IIS/11/20) over Rs 500'000.00, corresponding to USD 6730.00) from Cipla (https://www.cipla.com) to finance a part-time study coordinator for this trial. Cipla had no role in the trial design, data collection, data analysis, data interpretation, or writing of the manuscript.

**Competing interests:** The authors have declared that no competing interests exist.

doxycycline was associated with a relative risk reduction (RRR) and absolute risk reduction (ARR): ITT 31.6% RRR, 7.4% ARR (P = 0.063); PP 40.7% RRR, 9.6% ARR (P = 0.017); AT 43.2% RRR, 10.8% ARR (P = 0.007), with numbers needed to treat (NTT) of 13.4 (ITT), 10.4 (PP), and 9.3 (AT), respectively. Doxycycline was well tolerated with not a single patient stopping treatment due to adverse events.

## Conclusions

In hospitalized COVID-19 patients, doxycycline, a safe, inexpensive, and widely available antibiotic with anti-inflammatory properties, reduces the need for ICU admission when added to SoC.

## Introduction

The COVID-19 pandemic is an enormous burden on health care systems and economies around the globe. While effective vaccines directed against the SARS-CoV-2 spike protein are available, it will take much time and effort for worldwide immunization levels to be high enough to bring the COVID-19 pandemic to an end [1]. Variants of concern (VOC) of SARS-CoV-2 with increased transmissibility, such as the Delta (B.1.617.2) and, more recently, the Omicron (B.1.1.529) variants, have led to a substantial surge in the incidence rate of severe COVID-19 cases, particularly in the context of low vaccine coverage and especially in lower-middle and low-income countries. As a result, in many countries the saturation of intensive care units (ICU) has driven political decisions, including containment and segregation measures [2, 3].

A hyperinflammatory syndrome or cytokine storm contributes to severe disease [4], indicating a key role for dysregulated host innate immune mechanisms during advanced stages of COVID-19. Thus, major global efforts to develop effective treatments have targeted either the SARS-CoV-2 virus or the excessive inflammatory responses [5]. Conversely, the airway microbiome may also play a role in inflammatory responses, and bacterial superinfection of viral pneumonia is a well-known driver of severe disease and complications [6].

The development of new medicines and repurposing of existing drugs with potential value in COVID-19 has required large resources for the delivery of national and international trials. Complex networks have had to be set up, including the RECOVERY and PRINCIPLE platform trials in the UK and ACTIV in the USA, with additional international networks, such as the REMAP-CAP and the World Health Organisation (WHO) Solidarity Plus trial. This global effort has identified several effective treatments which feature in COVID-19 treatment guidelines [7]. Despite these advances, there remains a high unmet need for treatments that prevent COVID-19 progression in hospitalized patients to severe disease requiring transfer to an intensive care unit (ICU).

Doxycycline is an established, widely available and inexpensive oral drug with a good safety profile and pleiotropic therapeutic effects. It exerts broad-spectrum activity against both extra- and intracellular pathogens [8, 9], and is indicated for empiric treatment of community acquired pneumonia, including influenza-associated pneumonia [10, 11]. Doxycycline exhibits anti-inflammatory, anti-oxidative and tissue-protective effects, e.g. through potent inhibition of metalloproteinases, which are associated with the severity of COVID-19 [9, 12–14]. Therefore, doxycycline may be an effective treatment in hospitalized patients, who are at high risk for progressing to severe COVID-19 that requires treatment in ICU.

To test this hypothesis, we conducted a randomized, open-label, multi-centre trial based in 6 hospitals in India with high quality ICU facilities. The need for ICU admission was chosen as

the primary outcome, and the study was powered based on the assumption that, with standard of care treatments (SoC) at the time of protocol development for this trial, up to 25% of hospitalized COVID-19 patients require ICU admission [15]. The trial followed all the principles of good clinical practice (GCP) but differed from standard clinical trials in respect of its low total cost (< USD 10,000), including free of charge participation of the study team and all study sites) and the use of an online application (developed by JK and LG) serving as the electronic case report form (eCRF) that enabled efficient screening, randomization and collection of outcomes data.

## Methods

### Study design

This was a parallel group, controlled (against standard of care, SoC), randomized trial, including a screening period (0–1 days), 14 days of treatment and 14 days of follow-up by telephone. The study was conducted at six sites across India. Using the screening data entered by the attending physician(s) into the online app designed specifically for this trial (see data collection below), patients were randomized by the online app within 24 hours of admission to hospital into one of two arms: SoC or SoC and doxycycline (SoC+Doxy). The study was approved by ethics committees from Fortis Hospital (EC Ref No:220/EC/PI/2020) on 30.09.2020, and from CMRI (Ref Nr: IEC/01/2021/ACD-CT/APRV/05). All patients gave their written informed consent. The trial was registered on the Clinical Trials Registry–India (final registration number CTRI202105033867). The full study protocol is available at: https://cmri.ckbirlahospitals. com/Study_Protocol_DOXPREVENT.pdf (S1 File).

**Participants.**   We enrolled adult symptomatic patients with a proven diagnosis of COVID-19 who had been admitted to hospital within the last 24h. The following inclusion criteria had to be met: able to give informed consent, age ≥40 and <90 years, SARS-CoV-2 infection demonstrated by PCR, typical symptoms of COVID-19 (new onset or exacerbation of a pre-existing cough due to a chronic respiratory illness, dyspnea, increased body temperature (axillary T˚ > 37.6˚C or oral T˚>38˚C), and admission to hospital within 10 days of onset of symptoms. Exclusion criteria were: hypersensitivity to doxycycline; myasthenia gravis, pregnancy, and hepatic failure (CHILD-Pugh score C).

**Study endpoints.**   The question of scientific interest that the trial was designed to assess was whether treatment with doxycycline reduces the need (indication) for ICU care in newly hospitalized patients with symptomatic COVID-19. Therefore, the primary endpoint was the need for transfer to ICU within 14 days of admission, as judged by the attending physicians. Secondary endpoints included in the study protocol were: death, mechanical ventilation, time to discharge, recovery from symptoms, resolution of fever, prolonged hospital stay (>7days), and supplemental oxygen required.

**Study treatment.**   All patients received SoC set for their hospital and in keeping with the guidelines in India. Patients randomized to the SoC+Doxy arm were given doxycycline 100 mg BID for 14 days. Patients discharged before the 14-day period of follow up were contacted by telephone two days after discharge to ensure they are well and convalescing; those assigned to the SoC+Doxy arm were asked to complete 14 days of treatment with doxycycline in addition to any other drugs they were discharged with.

### Data collection and analysis

**Data collection with an online app.**   An electronic case report form (eCRF) app was developed (by JK and LG) for online entry (by the attending physicians), randomization, stratification, patient screening, treatment and outcomes data. The eCRF app was hosted on a

secure Amazon Web Services (AWS) server accessed by investigators via a user-specific user-name and password. The app consisted of 5 pages: screening, randomization, treatment, adverse events, and outcome. Instructions for its use (see supplemental appendix) were sent to all participants, and several online training sessions were held, including completion of mock patient data. All users had to be fully proficient before getting permission to enter real patient data. Specific instructions were issued for when the data were to be placed within each of the five pages.

The app was written in R Shiny and was hosted on shinyapps.io. Authentication based on user id and password was required to access the app. Data were stored in an AWS Aurora data-base instance, encrypted both at rest and in transit. It was only accessible via a secure, sepa-rately authenticated RestFUL API written in PHP and hosted on an AWS Elastic Compute server. All data entry and modification were audit trailed. The app downloaded and uploaded data to the database via the API securely via the SSL protocol. Data for analysis was also down-loaded via the API interface. R version 4.0.3 was used to perform all analyses of data collected in the study. The quality of the data was controlled by JK and all queries were passed on to the individual sites, assisted by the study coordinator based in India.

**Statistical analysis.** The primary endpoint was the need for patient transfer to ICU within 14 days of hospital admission as judged by the attending physicians. Based on avail-able data from the literature at the time of study planning in the first half of 2020, we assumed that up to 25% of patients admitted to hospital with COVID-19 would require ICU care within 14 days of admission [15]. The therapeutic effect size of doxycycline used for calculating the trial size was based on the pragmatic view that, in order to transform medical practice in hospitalized COVID-19 patients, addition of doxycycline to SoC should reduce the proportion of study participants meeting the primary outcome to 12.5%, a risk ratio of 0.500.

Initially, we planned a 2:1 (SoC+Doxy:SoC) randomization and interim analyses, allowing the possibility of stopping for both success and futility once the status of 50% and 75% of par-ticipants were known. In addition, we allowed for a futility-only interim once the status of 25% of participants was known. Loss to follow up was assumed to be ≤5%. The trial required 80% power to detect a reduction in the risk of needing admission to ICU from 25% to 12.5% using a one-sided significance level of 2.5%. These criteria gave a sample size of 347 in total, 231 in the doxycycline + SoC arm and 116 in the SoC arm. Due to an error during the implementa-tion of the randomisation, the app used a 1:1 allocation ratio. It should be noted that, for any given sample size, the power of a study using a 1:1 allocation ratio is greater than that of a study that uses any other allocation ratio. For full details of the initial and revised sample size calculations, see the supplemental appendix (S2 File).

**Stratification.** As the benefits of COVID-19 treatments may be different depending on prior co-morbidities, study participants were stratified as follows: no relevant prior illness; pre-existing lung conditions (ILD, COPD, bronchiectasis, asthma); and other relevant non-respiratory comorbidities (e.g. diabetes, heart disease, uncontrolled hypertension, cancer). Par-ticipants with both pulmonary and non-pulmonary co-morbidities were included in the pul-monary stratum.

Generalised linear models (GLMs) with a binomial response and canonical link were used to analyse the primary endpoint. The predictor variables used were stratum, sex and treatment. As no interim analyses were conducted, p-values quoted are two-sided rather than one-sided. For more details of statistical analyses, please see the supplemental appendix. Other baseline and outcome variables were reported using summary statistics appropriate for either categori-cal or continuous variables, as needed.

### In vitro studies

To test and compare the effects of doxycycline, tetracycline and remdesivir on SARS-CoV-2 infection of ACE-2-overexpressing lung-derived cell line A549-hACE2 and breast cancer cell line MDA-MB-231-hACE2, we carried out a standardized, luminometric viability assay following drug titration and exposure to six SARS-CoV-2 variants, in principle as reported [16]. For full details of the methods used see the supplemental appendix (S3 File).

### Study sponsorship and coordination

This was an investigator-led study without formal sponsorship as no institution (national or international) was identified for this purpose. The pharmaceutical company, Cipla (Mumbai, India) generously provided Rs 500,000 (USD 6730) as an unrestricted grant to support a part-time salary of a study coordinator who assisted the Chief Investigator (RDh). Cipla had no role in the trial design, data collection, data analysis, data interpretation, or writing of the manuscript. The costs related to the secure AWS server hosting the eCRF app (USD 1000) were borne by JK, RD and SDG. JK, RD, RDh, and SDG had full access to all the data at the end of the study, and all authors had individual responsibility for the decision to submit for publication. The corresponding author (SDG) had the final responsibility to submit the paper for publication.

## Results

### Study conduct

Enrollment into the trial started on 1st November 2020 and continued for 27 weeks until 10 May 2021 at 6 individual study sites across India. In total, 387 patients participated, of whom 228 (58.9%) were enrolled from mid-March until early May 2021, during the second COVID-19 peak in India (Figs 1 and 2) [17]. During this period, extremely high hospital admission rates of COVID-19 resulted in rapid recruitment into the trial, but also placed additional pressure on clinical staff at trial sites. This created gaps in communication between the steering committee and the study sites, leading to the unintentional omission of the pre-specified interim analysis. On 14 May 2021, after 387 patients had been enrolled, the trial steering committee formally confirmed that enrollment should stop. As the number of patients enrolled exceed the number needed for a conventional (not group sequential) design, and since the randomization ratio actually used was 1:1 rather than 2:1, the study had clearly achieved its intended power. The trial was terminated, enrolment ceased, ongoing patients were followed to completion, all data queries were resolved, and statistical analyses completed.

### Study participants

Of the 387 participants, 195 were randomized to the SoC arm and 192 to the SoC+Doxy arm. Baseline demographics were comparable between the two groups (Table 1). At study entry, all the participants had a positive PCR test for SARS-CoV-2 and were clinically diagnosed as having COVID-19. The median age (IQR, min, max) of participants was 58 (49–66, 40, 90) years, with a 2:1 ratio of males to females. The median axillary body temperature (IQR, min, max) at study entry was 38.2˚C (38.0˚C– 38.4˚C, 37.8˚C, 40.8˚C) in the SoC+Doxy group and 38.3˚ C (37.6˚C– 38.3˚C, 37.6˚C, 40.6˚C) in the SoC group.

Among the 387 study participants, 290 (74.9%) had at least 2 pre-existing comorbidities associated with increased risk for severe COVID-19, including hypertension, diabetes, heart disease, lung diseases, and cancer, while 97 participants (25.1%) had no identifiable risk factors at study entry. A pre-existing lung disease was present in 84 of 387 patients (21.7%). The incidence of various risk factors (RF) was similar in the two treatment groups (Table 1), with total

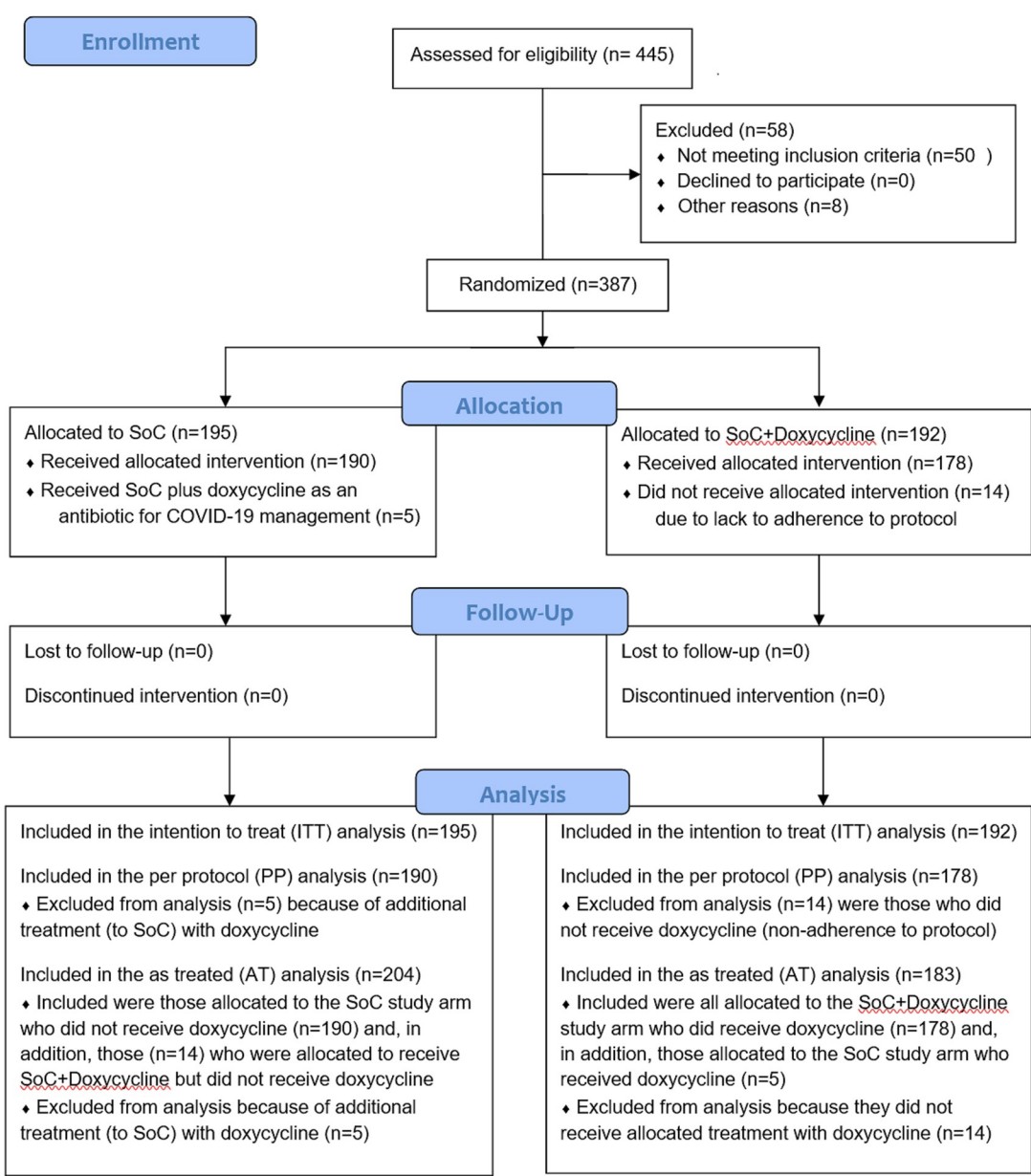

**Fig 1. Enrollment to analysis (CONSORT diagram).**

numbers of 579 RF in SoC and 570 in SoC+Doxy groups, and an average of 3 RF per study participant in both groups (S1 Table).

## Study treatment

Adherence to the protocol and study treatment per protocol was high, with 368 of 387 (95.1%) of participants receiving the allocated treatment based on randomisation. Of the 192 patients

(A)

(B)

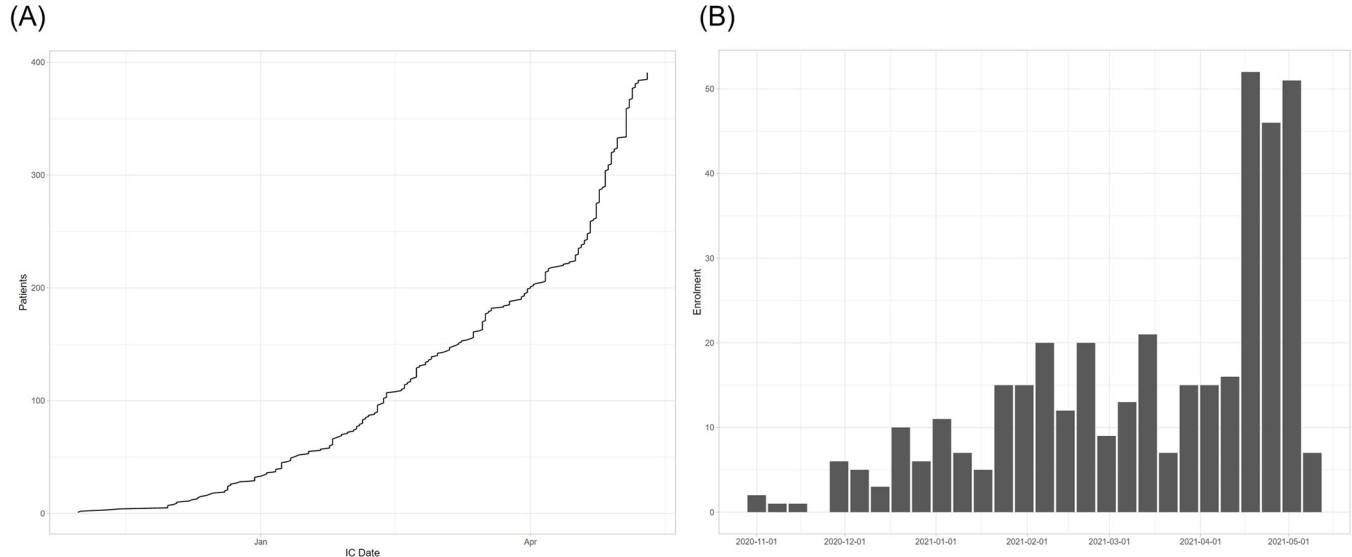

**Fig 2. Enrollment into the trial from 1st November 2020 until 10th May 2021.** (A) Enrollment over time. Cumulative number of enrolled patients shown on the y-axis. (B) Enrollment by week. Black columns represent one full week of enrollment.

randomized to the SoC+Doxy arm, 178 (92.7%) received doxycycline, while 5 of 195 patients (2.6%) randomized to the SoC arm were prescribed doxycycline as an antibiotic following the site's local guidelines for COVID-19 management (Table 1). Hence, 183 of the 387 (47.3%) patients in this trial were treated with SoC plus doxycycline, while 204 patients (52.7%) received treatment with SoC only.

Participants in the two treatment groups were similar with respect to concomitant medication (Table 1), with >80% of patients in both study arms receiving antibiotics other than doxycycline, glucocorticoids (dexamethasone or methylprednisolone), antivirals (remdesivir; favipiravir), and anticoagulants as SoC. Four patients in the SoC arm and none in the SoC +Doxy arm received tocilizumab. Other commonly used treatments included analgesics, anti-hypertensives, anti-diabetic drugs, and ivermectin.

## Primary outcome

Of the 387 patients included in the trial, 77 (19.9%) reached the primary outcome, i.e. they developed COVID-19 symptoms deemed by the attending physicians severe enough to require critical care in the ICU (Fig 3). Among the 290 patients with pre-existing comorbidities, 65 (22.4%) needed ICU admission, compared to only 12 of 97 patients without risk factors (12.4%) (OR 2.0463, 95% CI 1.0529 to 3.9768, p = 0.0347). The proportions requiring ICU treatment were similar in patients with pre-existing lung disease (18 of 84; 21.4%) and those with other risk factors (47 of 206; 22.8%).

For comparison of the primary outcome between the two treatment arms in this trial we carried out three types of analysis: First, in the intention to treat analysis (ITT) of all 387 participants, 46 of 195 patients (23.6%) randomized to SoC reached the primary outcome compared to 31 of 192 patients (16.1%) randomized to SoC+Doxy (OR 0.617, 95% CI 0.369 to 1.027, p = 0.063) (Fig 4). As 19 of the 387 study participants (4.9%) had received a treatment that was contrary to the randomization, two additional analyses were carried out for the primary outcome. First, a "per protocol" (PP) analysis included only the 368 patients who had received study treatment in accordance with their randomization; among these, 45 of 190 patients

**Table 1. Baseline characteristics and drug treatment during the study.**

| randomized (n) | All | Doxycyline group | SoC group |
|---|---|---|---|
| | **387** | **192** | **195** |
| Age, yrs (range) | 58.6 (40–90) | 58.6 (40–90) | 58.6 (40–88) |
| n Female:Male (%) | 140:247 (36.2:63.8) | 69:123 (36:64) | 71:124 (36.4:63.6) |
| n SARS-CoV-2 PCR+ (%) | 387 (100) | 192 (100) | 195 (100) |
| Body temperature (˚C), mean (range) | 38.3 (38–41) | 38.3 (38–41) | 38.4 (38–41) |
| **Comorbidities, n (%)** | | | |
| No comorbidity | 97 (25.1) | 48 (12.4) | 49 (12.7) |
| Any comorbidity | 290 (74.9) | | |
| Hypertension | 206 (53.2) | 101 (52.6) | 105 (53.9) |
| Diabetes | 138 (35.7) | 65 (33.9) | 73 (37.4) |
| Heart disease | 52 (13.4) | 28 (14.6) | 24 (12.3) |
| Lung disease (all) | 89 (23.0) | 45 (23.4) | 44 (22.6) |
| COPD | 35 (9.0) | 20 (10.4) | 15 (7.7) |
| Asthma | 29 (7.5) | 13 (6.8) | 16 (8.2) |
| ILD | 5 (1.3) | 3 (1.6) | 2 (1.0) |
| Bronchiectasis | 4 (1.0) | 2 (1.0) | 2 (1.0) |
| Other lung disease | 16 (4.1) | 7 (3.7) | 9 (4.6) |
| Cancer | 5 (1.3) | 2 (1.0) | 3 (1.5) |
| **Drug treatment during study, n (%)** | | | |
| Doxycycline | 183 (47.3) | 178 (92.7) | 5 (2.6) |
| Antibiotics other than doxycycline | 314 (81.1) | 158 (82.3) | 156 (80.0) |
| Glucocorticoids | 315 (81.4) | 160 (83.3) | 155 (79.5) |
| Antivirals | 355 (91.7) | 181 (94.3) | 174 (89.2) |
| Anticoagulants | 338 (87.3) | 174 (90.6) | 164 (84.1) |
| Antihypertensives | 200 (51.7) | 101 (52.6) | 99 (50.8) |
| Antidiabetic drugs | 144 (37.2) | 70 (36.4) | 74 (38.0) |
| Ivermectin | 152 (39.3) | 79 (41.2) | 73 (37.4) |
| Tocilizumab | 4 (1.3) | 0 (0) | 4 (2.1) |

(23.7%) in the SoC arm compared to 25 of 178 patients (14.0%) randomized to SoC+Doxy developed severe COVID-19 that met the criteria for ICU admission (OR 0.521, 95% CI 0.300 to 0.890, p = 0.017). Second, an "as treated" (AT) analysis of the primary outcome data was carried out after re-allocating the 19 patients to the two treatment arms by correcting for actual treatment. This showed that among the 183 participants who had actually received doxycycline in addition to SoC, a total of 26 of patients (14.2%) required ICU admission compared to 51 of the 204 (25.0%) participants treated with SoC only (OR 0.493, 95% CI 0.288 to 0.828, p = 0.007) (Figs 3 and 4). The observed effect of doxycycline on the requirement for ICU treatment was similar across different study sites, i.e., no batch effects were noted. Calculated relative risk reductions (RRR) were 31.6% (ITT), 40.7% (PP) and 43.2% (AT), with numbers needed to treat (NNT) ranging from 13.4 (ITT) to 9.3 (AT) (Table 2).

## Key secondary outcomes

Of the 77 patients judged by their physicians to require admission to the local ICU during the study, 46 (59.7%) were actually admitted, the main reason being shortage of ICU beds, especially during the second wave of the pandemic which started in mid-March 2021. In some cases, patients were deemed unsuitable for ICU admission based on low likelihood of surviving subsequent weaning from ventilation. 30 of 195 patients randomized to SoC and 16 of 192

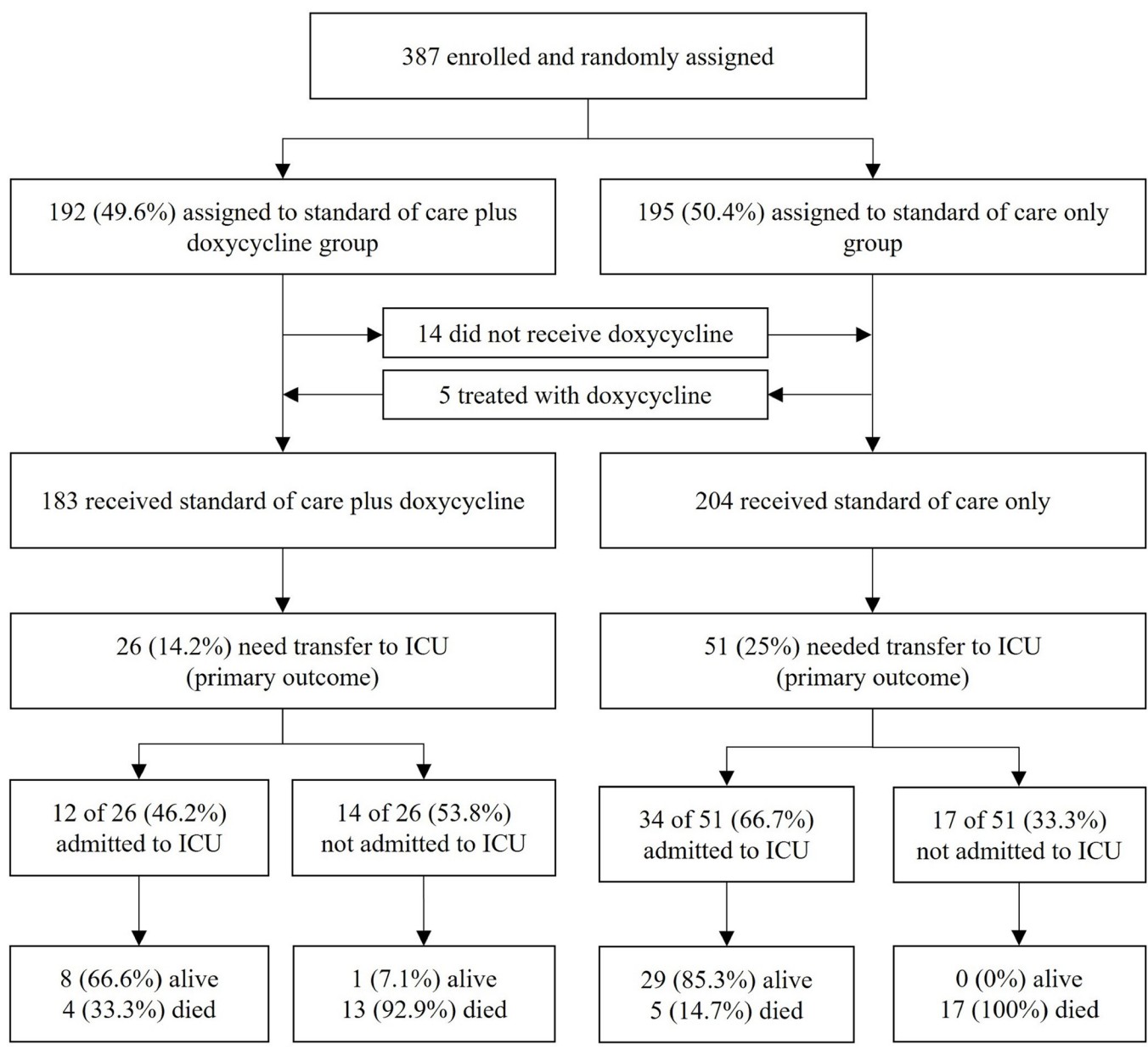

**Fig 3. Outcomes (CONSORT diagram).**

patients randomized to SoC+Doxy were actually admitted to ICU (OR 0.5000, 95% CI 0.2629 to 0.9510, p 0.0346). Among the 204 patients who were actually treated with SoC, 34 (16.7%) were admitted to ICU, compared to 12 of 183 (6.6%) patients treated with SoC plus doxycycline (OR 0.3509, 95% CI 0.1757 to 0.7006, p 0.003). Among patients requiring ICU treatment as judged by their treating physicians, 34 of 51 (66.7%) treated with SoC compared to 12 of 26 (46.2%) treated with SoC plus doxycycline were admitted to ICU (Fig 3). Thus, among patients requiring critical care, those treated with SoC plus doxycycline were less likely to be admitted to ICU than patients treated with SoC only, although the difference did not reach significance (OR for not being admitted to ICU 0.4286, 95% CI 0.1631 to 1.1262, p = 0.0856). We have no explanation for this observed difference in treatment.

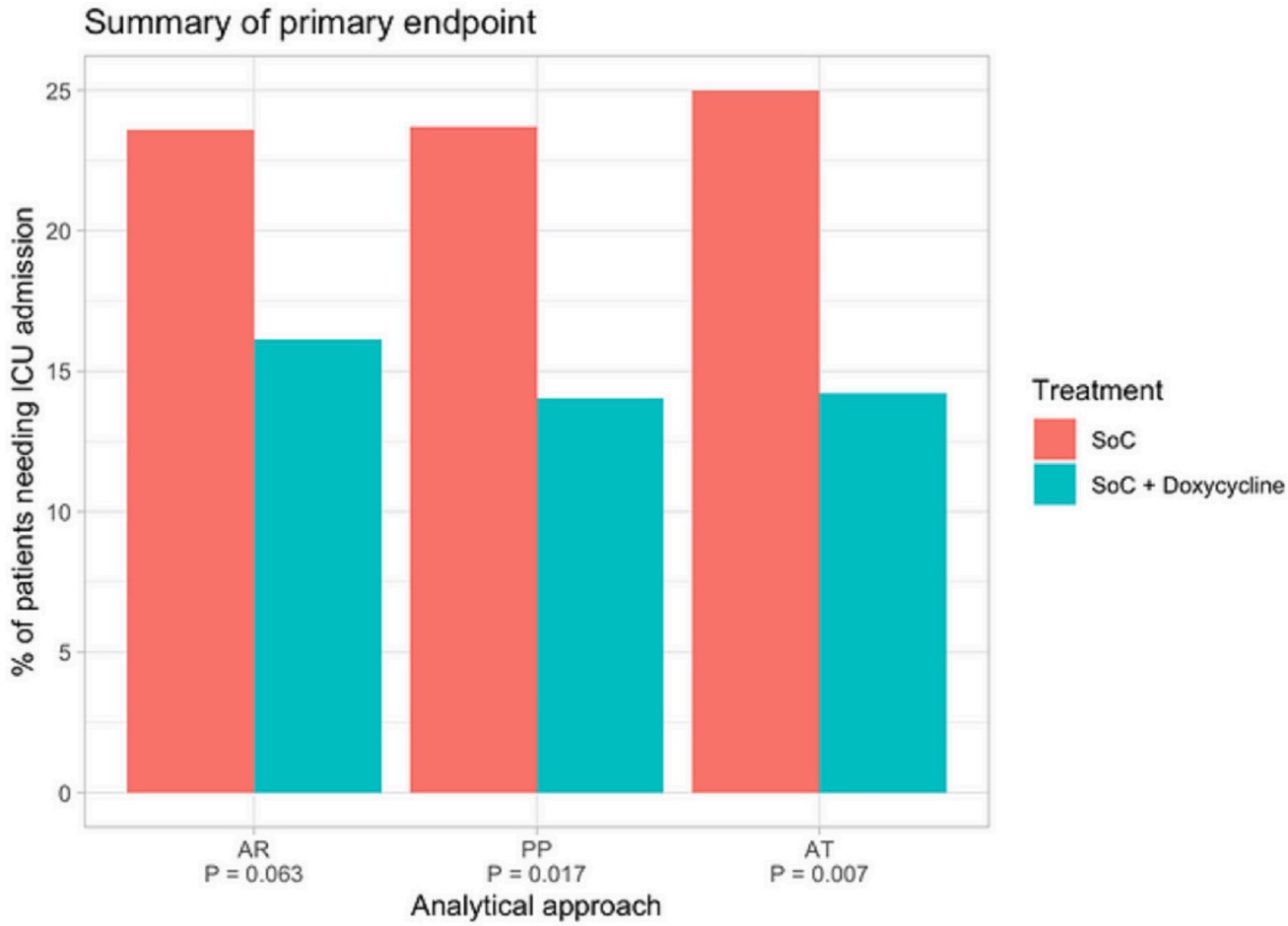

**Fig 4. Summary of primary outcome data.** Analyses of primary outcome data (need for ICU admission). ITT, as randomized; PP, per protocol; AT, as treated. Red bars: SoC; green bars: SoC+Doxy.

Thirty-nine of 387 study participants (10.1%) died during the trial. All deaths occurred in hospital. No deaths were recorded after discharge during the 14 day telephone follow-up period. Thirty of the 31 patients (96.8%) who required intensive care but could not be admitted to an ICU died, compared to 9 of 46 patients (19.6%) who did receive critical care in ICU (Fig 3)(OR 0.008, 95% CI 0.001 to 0.068, p<0.0001). Death was significantly associated with pre-existing comorbidities with 35 of 290 (12.1%) patients with pre-existing comorbidities (known to be associated with severe COVID-19) died, as compared with only 4 of 97 (4.1%) without

**Table 2. Relative Risk Reduction (RRR), Absolute Risk Reduction (ARR) and Number Needed to Treat (NNT) according to the three primary outcome analyses.**

| Analysis | SoC | SoC +Doxy | All | RRR (%) | ARR (%) | NNT (n) |
|---|---|---|---|---|---|---|
| **Intention to treat (ITT) (n)** | 195 | 192 | 387 | 31.6 | 7.4 | 13.4 |
| ICU required, n (%) | 46 (23.6) | 31 (16.1) | 77 (19.9) | | | |
| **Per protocol (PP) (n)** | 190 | 178 | 368 | 40.7 | 9.6 | 10.4 |
| ICU required, n (%) | 45 (23.7) | 25 (14.0) | 70 (19.0) | | | |
| **As treated (AT) (n)** | 204 | 183 | 387 | 43.2 | 10.8 | 9.3 |
| ICU required, n (%) | 51 (25.0) | 26 (14.2) | 77 (19.9) | | | |

comorbidities (OR 0.3121, 95% CI 0.1080 to 0.9022, p 0.0316). Overall, no difference in death rate was observed between the two treatment arms in the ITT analysis of all 387 patients. Nineteen of 192 patients randomized to SoC+Doxy (9.9%) and 20 of 195 patients (10.3%) randomized to SoC died (p 0.90; OR 0.961, 95% CI 0.4348 to 1.6506; p = 0.91). In the AT analysis based on actual treatment, 22 of 204 patients who received SoC (10.8%) and 17 of 183 patients (9.3%) who received SoC+Doxy died.

## Other outcomes

No serious adverse events (SAE) related to doxycycline were observed, and there were no cases of discontinuations of doxycycline or SoC due to adverse events (AE). The mean duration of hospital stay in both groups was 9 days (range 5–40 days).

## Discussion

The results of this randomized, controlled trial provide evidence which suggests that doxycycline reduces the need for ICU admission in hospitalized COVID-19 patients. The trial was carried out in challenging circumstances during the second wave of the COVID-19 pandemic in India defined by the rapid rise of the delta variant of SARS-CoV-2 [17]. All study participants had a positive SARS-CoV-2 PCR test, fever and additional clinical symptoms of COVID-19, and a high proportion of them had several risk factors for developing severe COVID-19, including diabetes in one third of patients. The consistency of the results in all three analytical approaches applied provides confidence that our conclusion is robust.

The main reasons for selecting doxycycline for this trial were its pleiotropic, anti-inflammatory and anti-microbial effects, robust safety profile and low cost. Approved in 1967, doxycycline shows minimal side effects [18], and is also safe in patients with severely impaired renal function [19]. The safety and tolerability in the current trial were excellent, with not a single patient stopping treatment because of an adverse event. Therefore, we conclude that the benefit:risk ratio of doxycycline in this population is highly positive.

No difference was seen in the secondary outcome measure of death between the two treatment arms. Possible reasons include the insufficient power of the trial to detect differences in mortality and the observed difference in ICU admission rates, which unintentionally disadvantaged patients treated with SoC plus doxycycline compared to those treated with SoC only.

This study suggests that the antimicrobial effects of doxycycline were not important in defining the need for ICU. Approximately 40% of patients were enrolled before and 60% after the start of the second wave in mid-March 2021 when the delta variant B.1.617.2 became dominant, suggesting that its therapeutic effect was independent of SARS-CoV-2 variants. Approximately 90% of patients in both study arms received proven antiviral drugs with activity against SARS-CoV-2. Our *in vitro* studies showed that neither doxycycline nor tetracycline had an inhibitory effects on different SARS-CoV-2 variants, including five variants of concern evaluated on two established ACE-2-overexpressing human cell lines (S1 Fig) [16]. As an antiviral positive control, the clinical RNA-dependent RNA polymerase (RdRp) inhibitor remdesivir potently blocked SARS-CoV-2 replication and virus-induced cell death (S1 Fig). While studies by one laboratory in African green monkey-derived Vero-E6 cells suggested an antiviral effect of doxycycline *in vitro* [20, 21], another report has challenged this conclusion [22]. Potentially, the non-human origin of Vero-E6 cells may have been a confounder in the evaluation of anti-SARS-CoV-2-specific effects of doxycycline.

The UK platform trial, PRINCIPLE, to which some members of our study team contributed (SDG, JK, RD), found that doxycycline was not associated with reduced hospital admissions or death when used in the community in patients with suspected early COVID-19 [23]. This

contrast in benefits of doxycycline with the current study of advanced hospital stages of COVID-19 further supports the notion that doxycycline has no direct antiviral activity against SARS-CoV-2. Of note, dependency on the stage of disease is also observed with monoclonal antibodies against the SARS-CoV-2 spike protein which act directly as anti-viral agents and are effective during early disease [24]. A high proportion of patients in both treatment groups received broad spectrum antibiotics, which argues against an important contribution of antibiotic properties of doxycycline to the observed benefit. Of note, current guidelines do not recommend antibiotics for COVID-19 unless there is a clinical suspicion of bacterial co-infection [17].

Severe COVID-19 that requires critical care is associated with a hyperinflammatory state [4], which provides the pathobiological substrate for glucocorticoids that have been shown to be effective in COVID-19 patients requiring oxygen treatment. The fact that approximately 80% of patients in both treatment groups of this trial received glucocorticoids suggests that doxycycline acts via different mechanisms. Doxycycline has well described anti-inflammatory effects in various human lung diseases, including cystic fibrosis [25], lung fibrosis [26], and sarcoidosis [27]. In patients with Dengue hemorrhagic fever and hyperinflammation, doxycycline substantially reduces mortality in association with a significant reduction in serum concentrations of interleukin-6, TNFα and interleukin-1 [28]. Various pathways promoting inflammation and oxidative cell stress are targeted by doxycycline, including mitogen-activated protein kinase (MAPK) and Smad pathways, matrix metalloproteinases (MMP) implicated in inflammatory lung injury, and malondialdehyde-acetaldehyde (MAA) activation of Nrf2 [9, 12, 13]. In addition, doxycycline has been shown to directly scavenge reactive oxygen species, such as superoxide [13], and to reduce the production of nitric oxide [29], both of which have been implicated in lung injury and endothelial dysfunction [30]. Hence, the beneficial effects of doxycycline observed in this study may have been related to its anti-inflammatory, anti-oxidative, and cell-protective properties.

In addition to its potential pharmacodynamic effects and safety profile, the pharmacokinetic (PK) properties of doxycycline made it a good choice for this trial, especially its high bioavailability after oral dosing, short time required to achieve effective blood levels, a half-life of 12-25h, and strong tissue penetration into respiratory tissues, with reach into the intracellular space [8]. Of note, an oral dose of 200mg doxycycline achieves sufficient concentrations in sputum to inhibit MMP-9 [31].

The selection of doxycycline for this trial was also strongly driven by our intent to identify cost-effective therapies that would benefit patients across the globe and not just those in affluent countries which can afford the effective, yet very expensive, drugs for which efficacy against COVID-19 has been reported so far. In the UK, a 14-day course of 200 mg doxycycline costs £ 6.50, while in India it is less than £5.00. The observed reduction in ICU admissions, when extrapolated to the large numbers of patients progressing to severe forms of COVID-19, also has the potential for major cost-savings. This is especially relevant in countries like India, where the cost of ICU–based care provided by private sector hospitals is estimated at $255 per day [32]. While this may not seem excessive and given that universal healthcare is guaranteed by the Indian constitution, the limited number of tertiary government-run teaching hospitals means that much of the ICU care is provided by the private sector for which costs are met by patients and their families. Inevitably, those patients and families without financial means to bear such costs may have no other choice but to decline ICU admission despite a strong clinical need. Doxycycline is produced by many companies and is included in the WHO 2019 core list of essential medicines [33], so there should be sufficient stocked drug available in case it becomes a standard treatment for COVID-19 in hospitalized patients.

This study has also shown that randomized controlled trials can be done at minimal cost during a pandemic, and it provides proof of concept for the use of an online app as a means of

reliably collecting data in multi-centre trials. The only unavoidable, but ultimately trivial, cost was that of the server which enabled secure access to the application that served as an online eCRF. We sought help from a large Indian pharmaceutical company (Cipla) which generously provided a small unrestricted grant to support a research coordinator to help initiate the study by facilitating communication. None of the investigators or their institutions sought any compensation for the time and resources used. The study and the approach we took had its limitations, the main being the absence of the intensity of monitoring that drug trials usually entail. However, as a real-world study, this trial was just as determined and able as the major COVID-19 platform trials to obtain accurate data. We also recognize that we could not undertake the planned interim analyses because of strained communications during the second peak in India, something that may have been easier to cope with had more centres been recruited (meaning less burden per centre) and/or had trial monitors been assigned to each site, albeit raising the cost of the trial. Whilst we were confident about the main outcome (need for ICU admission) and mortality, we were less confident about collecting some of the secondary endpoints, so did not feel it justified to analyse these outputs. We conclude, therefore, that the approach taken, including the use of the eCRF, is less suited for collection of finer study details, although adjustments to improve the functionality of the app should make it possible to collect more complex datasets.

In summary, this study has shown that doxycycline prevents progression of COVID-19 to the point where ICU admission is needed. This comes with significant implications for both patients and health services, in particular in countries where, because of limited financial resources, ICU admission is dependent on accessibility and affordability rather than clinical indication. A similar study is needed to seek evidence of treatment on mortality, requiring large numbers of patients similar to those enrolled in well-funded, platform trials.

## Supporting information

**S1 Table. Number of Risk factors associated with severe COVID-19 in the study.**
(DOCX)

**S1 Fig. Antiviral effect of doxycycline, tetracycline and remdesivir.** Effect of doxycycline and tetracycline compared to remdesivir on SARS-COV-2 infection of ACE-2-overexpressing lung-derived cell lines *in vitro*.
(TIF)

**S1 File. Study protocol.**
(PDF)

**S2 File. Sample size calculation.**
(PDF)

**S3 File. Detailed material and methods of *in vitro* studies.**
(PDF)

**S4 File. Reporting checklist for randomised trials (based on the CONSORT guidelines).**
(PDF)

## Acknowledgments

We wish to acknowledge the following clinicians who, as members of the clinical teams attending to the patient participants on this trial, contributed to clinical management of the patients and data collection:

Dr Dipabali Acharjee, Research Associate, CMRI hospital, Kolkata, India.

Dr Virender Pratibh Prasad, Consultant Pulmonary Medicine, Yashoda Hospital, Somaji-guda, India.

Dr Aditya K. Chawla, Junior Consultant, Saroj Superspeciality Hospital and Jaipur Golden Hospital, Delhi and ii. Dr Gaurav Chaudhary, DNB Resident (Final yr), Jaipur Golden Hospital, Delhi, India.

Dr Vishnu Gireesh, Resident, Department of Chest Medicine, Ruby Hall Clinic, Pune, India.

Dr Avhinav Bhosle, Consultant Pulmonologist, Kalyan Hospital, Vadodara, India.

Dr Viswesvaran Balasubramanian, Lead Consultant and Head, Department of Pulmonary and Sleep Medicine, Yashoda hospitals, Hyderabad

## Author Contributions

**Conceptualization:** Ratko Djukanovic, Stephan D. Gadola.

**Data curation:** John Kirkpatrick, Laura Gilbert.

**Formal analysis:** Raja Dhar, John Kirkpatrick, Laura Gilbert, Marcel Stern, Oliver T. Keppler, Ratko Djukanovic, Stephan D. Gadola.

**Funding acquisition:** Raja Dhar.

**Investigation:** Raja Dhar, Arjun Khanna, Mahavir Madhavdas Modi, Rakesh K. Chawla, Sonia Dalal, Venkata Nagarjuna Maturu, Marcel Stern, Oliver T. Keppler, Stephan D. Gadola.

**Methodology:** John Kirkpatrick, Laura Gilbert, Marcel Stern, Oliver T. Keppler, Ratko Djukanovic, Stephan D. Gadola.

**Project administration:** Raja Dhar, John Kirkpatrick, Ratko Djukanovic.

**Software:** John Kirkpatrick, Laura Gilbert.

**Supervision:** Raja Dhar, John Kirkpatrick, Ratko Djukanovic, Stephan D. Gadola.

**Writing – original draft:** Raja Dhar, John Kirkpatrick, Marcel Stern, Oliver T. Keppler, Ratko Djukanovic, Stephan D. Gadola.

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
