## [Decision Letter · Decision Letter 0]

8 Sep 2022

PONE-D-22-10587Doxycycline for the prevention of progression of COVID-19 to severe disease requiring intensive care unit (ICU) admission: a randomized, controlled, open-label, parallel group trial (DOXPREVENT.ICU)PLOS ONE

Dear Dr. Gadola,

Thank you for submitting your manuscript to PLOS ONE. After careful consideration, we feel that it has merit but does not fully meet PLOS ONE’s publication criteria as it currently stands. Therefore, we invite you to submit a revised version of the manuscript that addresses the points raised during the review process.

ACADEMIC EDITOR: Please make revisions as suggested by reviewers of write a detailed rebuttal on a point-by-point basis.

We look forward to receiving your revised manuscript.

Kind regards,

Davor Plavec, MD, MSc, PhD, Prof.

Academic Editor

PLOS ONE

Journal Requirements:

Additional Editor Comments:

Dear Authors,

please make revisions as suggested by reviewers of write a detailed rebuttal on a point-by-point basis.

Reviewers' comments:

Reviewer's Responses to Questions

**Comments to the Author**

1. Is the manuscript technically sound, and do the data support the conclusions?

Reviewer #1: Yes

Reviewer #2: Yes

2. Has the statistical analysis been performed appropriately and rigorously? 

Reviewer #1: Yes

Reviewer #2: No

3. Have the authors made all data underlying the findings in their manuscript fully available?

Reviewer #1: Yes

Reviewer #2: Yes

4. Is the manuscript presented in an intelligible fashion and written in standard English?

Reviewer #1: Yes

Reviewer #2: Yes

5. Review Comments to the Author

Reviewer #1: 1. PRINCIPLE, UK platform found that doxycycline was not associated with reduced hospital admission or death when used in the community in patients with suspected early COVID-19.

No difference was seen in the secondary outcome measure of death between the two treatment arms in this study. Also, results presented at the American College of Allergy, Asthma & Immunology (ACAAI) 2021 Annual Scientific Meeting are similar. Use of doxycycline is not associated with improved mortality rate in patients with COVID-19 pneumonia. How to explain these results?

2. Authors mentioned that doxycycline has no in vitro activity against SARS-CoV-2. This is in contrast with findings from other authors (e.g. Gendrot).

Reviewer #2: Mortality is not clearly defined. Is it measured at discharge or in follow-up? A survival analysis may be more suitable.

Most secondary outcomes were not analyzed or presented in the results section.

The results section was not clearly written. The primary analysis should focus on the comparison between two treatment groups. However, in “Primary Outcome”, primary findings were buried in subgroup analysis (Eg. Comorbidities, lung disease and etc). The subgroup analysis should be moved to another section. In “Key secondary Outcomes”, ICU admission were mentioned again. This is confusing as ICU is not secondary but primary.

Table 1 present either mean with SD or median with range. How are the two groups compared?

Add p values to Table 1 and 2.

Table 2 put n (%) in one cell.

Figure 2 a flow chart for outcome is not necessary and can be omitted.

“The two treatment groups were matched” on Page 7 study participants and Page 8 2nd paragraph. The authors may change the statement to “two groups are not significantly different”. Matching has different meaning in statistics.

6. PLOS authors have the option to publish the peer review history of their article (what does this mean?). If published, this will include your full peer review and any attached files.

Reviewer #1: No

Reviewer #2: No

---

## [Author Response · Author response to Decision Letter 0]

29 Oct 2022

As requested by the academic editor, a rebuttal letter that responds to each point raised by the academic editor and the two reviewers of our manuscript has been uploaded as a separate file labeled 'Response to Reviewers'.

The responses contained in our letter are reproduced in the following:

II. Responses to reviewers’ comments on general aspects of the manuscript.

1) Is the manuscript technically sound, and do the data support the conclusions?

Response: We thank both reviewers for agreeing that our manuscript is technically sound and that the data support the conclusions.

2) Has the statistical analysis been performed appropriately and rigorously?

Response: Reviewer #2 raised a couple of points in relation to the statistical analysis. We will address these points below (in the section « 5. Response to reviewers’ comments on specific aspects of the manuscript »).

3) Have the authors made all data underlying the findings in their manuscript fully available?

Response: We thank both reviewers for appreciating our effort to make all data underlying our findings available to the public.

4) Is the manuscript presented in an intelligible fashion and written in standard English?

Response: Thank you for appreciating the clarity and writing style of our manuscript. 

III. Point-by-point responses to reviewers’ comments on specific aspects of the manuscript.

Points raised by Reviewer #1

1) PRINCIPLE, UK platform found that doxycycline was not associated with reduced hospital admission or death when used in the community in patients with suspected early COVID-19.

Response: The main differences between the trial reported in the manuscript under consideration and that reported by the PRINCIPLE consortium, to which three of us contributed as co-authors, relate to the patient population enrolled in the respective trials. While PRINCIPLE was a community-based trial for all-comers from age 18 onwards, therefore including mostly people with mild COVID-19, our trial was designed to test the effect of doxycycline in patients who were severe enough to require hospital admission. Thus, the patients in the current trial had more advanced disease stage COVID-19 and were, therefore, at a much higher risk for severe outcome. Of note, during the 2nd wave of the pandemic in India, hospitals were at their capacity and stretched to their limit and only severely ill patients could be admitted. In fact, the situation was so dramatic that many COVID-19 patients died outside of these hospitals before they could be admitted). The differences in population between our trial and PRINCIPLE are evident from the very different (almost 20-fold different) death rate between the two trials, i.e. 0.6% death rate in PRINCIPLE versus 10.1% death rate in our trial. In conclusion, we argue that these two trials cannot be directly compared.

To emphasize this point more clearly in the manuscript, we have revised the text of the manuscript as follows: 

Section: “Discussion”; 5th paragraph: «The UK platform trial, PRINCIPLE, to which some members of our study team contributed (SG, JK, RD), found that doxycycline was not associated with reduced hospital admissions or death when used in the community in patients with suspected early COVID-19.23 Of note, the population of this trial was much more severe than that studied in PRINCIPLE, a community-based platform trial. While most patients enrolled in PRINCIPLE had mild disease, all patients in this trial were severe enough to warrant hospital admission. This contrast in benefits of doxycycline between the PRINCIPLE and the current study of advanced hospital stages of COVID-19 further supports the notion that doxycycline has no direct antiviral activity against SARS-CoV-2. Of note, dependency on the stage of disease is also observed with monoclonal antibodies against the SARS-CoV-2 spike protein which act directly as anti-viral agents and are effective during early disease.20»

2) No difference was seen in the secondary outcome measure of death between the two treatment arms in this study. 

Response: We have discussed this in the manuscript (Discussion, 3rd paragraph), based on the results described in the section “Key secondary outcomes”. Due to space constraints and the fact that it is a secondary outcome measure, we kept this part of the discussion brief. However, if the reviewer insists and the editor agrees, we would be happy to expand in a further revision. In our view, two main aspects of this trial may explain why no statistically significant difference in death rate was observed between the two treatment arms:

1. Our prospective randomised clinical trial was not powered to see differences in death rate between the two study arms: A retrospective power calculation, based on the observed death rate of 10% in the SoC group, shows that a sample size of 868 patients (i.e. more than 2x the number of patients included in our trial) would have been required to observe (with alpha 0.05 and power 80%) a (quite dramatic) 50% reduction of the death rate. In order to observe a lesser, albeit still highly relevant 25% reduction of the death rate, a sample size of 4008 patients (i.e. >10x the number of patients in our trial) would have been required. A trial of that size was simply beyond our means.

2. Differences in ICU admission rates for patients of the two study arms who reached the primary outcome of “need for ICU admission”: 

A notable result of our study was that all but 1 of the 31 patients, judged by their doctors to require ICU treatment but could not be admitted to ICU, died (96.8%), while only 1 of 5 patients (9 of 46, 19.6%) admitted to ICU died. Thus, those patients in our trial whose COVID-19 deteriorated to the extent that they reached the primary outcome “need for ICU treatment” and could be admitted to ICU stood a much better chance of survival than those who reached the primary outcome but were not admitted to ICU for reasons that we explain in the paper. 

By chance (for unknown reasons), only 46% (12 of 26) of patients in the Doxycyline arm who reached the primary outcome were actually admitted to ICU, compared to 67% (34 of 51) of patients in the SoC arm. This bias in ICU admission rates, for which we have no explanation, may have confounded the secondary outcome of death in our study.

3) Results presented at the American College of Allergy, Asthma & Immunology (ACAAI) 2021 Annual Scientific Meeting are similar (to results of the PRINCIPLE trial). Use of doxycycline is not associated with improved mortality rate in patients with COVID-19 pneumonia. How to explain these results? 

Response: The study presented at the American College of Allergy, Asthma & Immunology (ACAAI) 2021 Annual Scientific Meeting was a retrospective chart study in 110 patients with COVID-19 pneumonia. The limited information provided in the brief study abstract does not allow us to assess the patient population or other relevant aspects of the study. It is, therefore, not possible to conduct a fair comparison between this the ACAAI study and our prospective randomised controlled trial. 

4) Authors mentioned that doxycycline has no in vitro activity against SARS-CoV-2. This is in contrast with findings from other authors (e.g. Gendrot).

Response: Thank you for raising this difference between our own in vitro study and that of Gendrot et al. For additional clarity, we have added text in the methods section of the main manuscript, adding an appropriate reference, and have expanded the discussion on previous in vitro studies, also adding three references. 

(Both the numbering of references and the list of references were updated accordingly).

Please note the following edits in the revised manuscript:

Section: “Methods”; subsection “In vitro studies”, 1st sentence: «To test and compare the effects of doxycycline, tetracycline and remdesivir on SARS-CoV-2 infection of ACE-2-overexpressing lung-derived cell line A549-hACE2 and breast cancer cell line MDA-MB-231-hACE2, we carried out a standardized, luminometric viability assay following drug titration and exposure to six SARS-CoV-2 variants, in principle as reported.16»

Section: “Discussion”, 4th paragraph, edited sentences 4 and new sentences 5-7: «Our in vitro studies showed that neither doxycycline nor tetracycline had an inhibitory effect on different SARS-CoV-2 variants, including five variants of concern evaluated on two established ACE-2-overexpressing human cell lines (PMID: 35090165) (Suppl. Figure 1). As an antiviral positive control, the clinical RNA-dependent RNA polymerase (RdRp) inhibitor remdesivir potently blocked SARS-CoV-2 replication and virus-induced cell death (Suppl. Figure 1).16 While studies by one laboratory in African green monkey-derived Vero-E6 cells suggested an antiviral effect of doxycycline in vitro,20, 21, a report by another group has challenged this conclusion.22 Potentially, the non-human origin of Vero-E6 cells may have been a confounder in the evaluation of anti-SARS-CoV-2-specific effects of doxycycline.»

Points raised by Reviewer #2: 

1) Mortality is not clearly defined. Is it measured at discharge or in follow-up? A survival analysis may be more suitable.

Response: Thank you for seeking more clarification. Survival analysis is normally conducted on data for which the period of observation is long. Given the relatively short duration over which the primary outcome was assessed (14 days) we did not, and still do not, think that a time-to-event analysis would add any useful insight. 

We are grateful for the question about the time-point at which mortality was measured and have, therefore, edited the text as follows:

Section: “Discussion”; subsection “Key secondary Outcomes”, New 2nd sentence in 2nd paragraph: «All deaths occurred in hospital. No deaths were recorded after discharge during the 14-day telephone follow-up period.»

2) Most secondary outcomes were not analyzed or presented in the results section.

Response: In this paper we have focused on the primary outcome “need for ICU admission”. The secondary outcome data are available for independent analysis via the supplementary information. However, any analysts seeking to undertake this should be aware that data collection for some secondary endpoints was not as complete as we had hoped due to pressures caused by the pandemic wave that India was going through at the time.

3) The results section was not clearly written. The primary analysis should focus on the comparison between two treatment groups. However, in “Primary Outcome”, primary findings were buried in subgroup analysis (Eg. Comorbidities, lung disease and etc). The subgroup analysis should be moved to another section.

Response: 

We agree with the reviewer that the clarity of this section can be improved. We have, therefore, edited the text in the “Primary Outcome” section as shown below.

We also agree with the reviewer that the focus of the primary analysis must be on the comparison between treatment groups, although we argue that we have actually built the section in this way. The first, introductory paragraph of the “Primary Outcome” section provides the relevant primary outcome data of the whole trial population and of the clinically important strata. As these are actual primary outcome data, we argue they should stay in this (“Primary Outcome”) section of the manuscript. On the other hand, to put them at the end of this section, which (in the actual version) ends with the most exciting result of the trial, seems inappropriate.

Edited text as follows:

Section: Results; subsection “Primary Outcome”, 1st paragraph, first sentence: 

«Of the 387 patients included in the trial, 77 (19.9%) reached the primary outcome, i.e., they developed COVID-19 symptoms deemed by the attending physicians severe enough to require critical care in the ICU (Figure 3).»

Section: Results; subsection “Primary Outcome”, 2nd paragraph, a new, first sentence and a slightly modified, second sentence as follows: 

«For comparison of the primary outcome between the two treatment arms in this trial we carried out three types of analysis: First, in the intention to treat analysis (ITT) of all 387 participants, 46 of 195 patients (23.6%) randomized to SoC reached the primary outcome compared to 31 of 192 patients (16.1%) randomized to SoC+Doxy (OR 0.617, 95% CI 0.369 to 1.027, p=0.063) (Figure 4).»

4) In “Key secondary Outcomes”, ICU admission were mentioned again. This is confusing as ICU is not secondary but primary.

Response: Thank you for giving us the opportunity to clarify. The primary endpoint of our trial is need for ICU admission. However, as explained in the manuscript, not all patients reaching this (primary) endpoint could actually be admitted to ICU, for reasons explained in the manuscript. Thus, actual ICU admission was included as a secondary endpoint and is discussed under key secondary outcomes in the manuscript. The reference of the primary outcome of need for ICU admission is to contextualise the secondary (but closely related) endpoint of actual admission.

5) Table 1 present either mean with SD or median with range. How are the two groups compared?

Response: We present means and ranges for continuous variables and n (%) for categorical ones. We can amend to mean (SD) if the editor/reviewers consider this essential for acceptance, but we feel the current presentations provide an equivalent level of detail. Interested readers have access to the raw data via the paper’s supplements if they are sufficiently motivated to derive alternative summaries of their own.

6) Add p values to Table 1 and 2.

Response: We respectfully disagree with the suggestion from reviewer 2 that p-values should be added to this table (and to table 2). Such an action is contrary to guidance from the CONSORT Consortium which states that "such significance tests assess the probability that observed baseline differences could have occurred by chance; however, we already know that any differences are caused by chance. Tests of baseline differences are not necessarily wrong, just illogical" 

(for full guidance please see https://www.consort-statement.org/checklists/view/32--consort-2010/510-baseline-data, second para) See also deBoer et al (2015) Int J Behav Nutr Phys Act https://doi.org/10.1186%2Fs12966-015-0162-z

7) Table 2 put n (%) in one cell.

Response: Table 2 has been edited according to the reviewer’s suggestion (please see revised manuscript).

8) Figure 2 a flow chart for outcome is not necessary and can be omitted.

Response: We believe Reviewer #2 is actually referring to Figure 3, as Figure 1 is the CONSORT diagram which needs to be included in the manuscript as per guidance of the journal, and Figure 2 is showing enrollment into the trial. The flow chart in Figure 3 provides the reader with an overview of the main results. We agree that it could be omitted from the main manuscript text and propose (to the academic editor) that, if he so chooses, the current Figure 3 can be moved to the supplementary section. 

9) “The two treatment groups were matched” on Page 7 study participants and Page 8 2nd paragraph. The authors may change the statement to “two groups are not significantly different”. Matching has different meaning in statistics.

Response: We thank the editor for this comment and have edited, accordingly, the text in the manuscript.

Please see the following edited text in the manuscript:

Section: “Results”, subsection “Study participants”, 2nd paragraph, third sentence, as follows: 

«The incidence of various risk factors (RF) was similar in the two treatment groups (Table 1).»

Section: “Results”; subsection “Study treatment”, 2nd paragraph, first sentence, as follows: 

«Participants in the two treatment groups were similar with respect to concomitant medication (Table 1), …»

Final note to academic editor

On review of our manuscript, we have deleted the following sentence from the manuscript as it is redundant:

Section: “Methods”, subsection “Data collection and analysis”, 2nd paragraph, deleted the following (second) sentence «Data could not be persisted on the app and, therefore, needed to be downloaded on every load, and new data needed to be uploaded before exit; thus, the app synchronized all changes in real time with a remote database in the cloud.» 

Thank You for considering our revised manuscript for publication in PLoS One.

---

## [Decision Letter · Decision Letter 1]

15 Nov 2022

PONE-D-22-10587R1Doxycycline for the prevention of progression of COVID-19 to severe disease requiring intensive care unit (ICU) admission: a randomized, controlled, open-label, parallel group trial (DOXPREVENT.ICU)PLOS ONE

Dear Dr. Gadola,

Thank you for submitting your manuscript to PLOS ONE. After careful consideration, we feel that it has merit but does not fully meet PLOS ONE’s publication criteria as it currently stands. Therefore, we invite you to submit a revised version of the manuscript that addresses the points raised during the review process.

ACADEMIC EDITOR: Please revise as suggested by the reviewers.==============================

We look forward to receiving your revised manuscript.

Kind regards,

Davor Plavec, MD, MSc, PhD, Prof.

Academic Editor

PLOS ONE

Journal Requirements:

Additional Editor Comments:

Please revise as suggested by the reviewer.

Reviewers' comments:

Reviewer's Responses to Questions

**Comments to the Author**

1. If the authors have adequately addressed your comments raised in a previous round of review and you feel that this manuscript is now acceptable for publication, you may indicate that here to bypass the “Comments to the Author” section, enter your conflict of interest statement in the “Confidential to Editor” section, and submit your "Accept" recommendation.

Reviewer #1: All comments have been addressed

Reviewer #2: (No Response)

2. Is the manuscript technically sound, and do the data support the conclusions?

Reviewer #1: Yes

Reviewer #2: (No Response)

3. Has the statistical analysis been performed appropriately and rigorously? 

Reviewer #1: Yes

Reviewer #2: (No Response)

4. Have the authors made all data underlying the findings in their manuscript fully available?

Reviewer #1: Yes

Reviewer #2: (No Response)

5. Is the manuscript presented in an intelligible fashion and written in standard English?

Reviewer #1: Yes

Reviewer #2: (No Response)

6. Review Comments to the Author

Reviewer #1: Thank you for your response. In my opinion, all major and minor comments have been adequately addressed.

Reviewer #2: For table 1, please follow the PLOS ONE guidelines

https://www.equator-network.org/wp-content/uploads/2013/03/SAMPL-Guidelines-3-13-13.pdf

Summarize data that are approximately normally

distributed with means and standard deviations

(SD). Use the form: mean (SD), not mean ± SD.

Summarize data that are not normally distributed

with medians and interpercentile ranges, ranges, or

both. Report the upper and lower boundaries of

interpercentile ranges and the minimum and

maximum values of ranges, not just the size of the

range

7. PLOS authors have the option to publish the peer review history of their article (what does this mean?). If published, this will include your full peer review and any attached files.

Reviewer #1: No

Reviewer #2: No

---

## [Author Response · Author response to Decision Letter 1]

18 Dec 2022

Response to Reviewer 2 

PONE-D-22-10587

Doxycycline for the prevention of progression of COVID-19 to severe disease requiring intensive care unit (ICU) admission: a randomized, controlled, open-label, parallel group trial (DOXPREVENT.ICU)

Dear Editor and reviewers,

We thank Reviewer #1 for his conclusion that all major and minor comments have been adequately addressed in our previously submitted revised manuscript.

Reviewer #2 raised the following point to which we will respond in the following.

Points raised by Reviewer #2 (in response to submitted 1st version of our revised manuscript): 

1) For table 1, please follow the PLOS ONE guidelines

(https://www.equator-network.org/wp-content/uploads/2013/03/SAMPL-Guidelines-3-13-13.pdf):

Summarize data that are approximately normall distributed with means and standard deviations (SD). Use the form: mean (SD), not mean ± SD.

Summarize data that are not normally distributed with medians and interpercentile ranges, ranges, or both. Report the upper and lower boundaries of interpercentile ranges and the minimum and maximum values of ranges, not just the size of the range 

Response: As it is unusual, in our experience, to distinguish between normal and non-Normal data rather than continuous and discrete data, it would be helpful if Reviewer 2 could indicate which continuous variables they regard as “approximately normally distributed” and those which they do not.

We are also unsure about the comment of Reviewer 2 in relation to “interpercentile range” without specification of the percentiles that define the lower and upper boundaries of the range. Reporting interquartile ranges as well as minima, maxima and medians makes no sense for dichotomous and other categorical variables (which are by definition not Normal), such as gender, baseline PCR test results, the presence or absence of comorbidities and the use of concomitant medications.

In an effort to oblige Reviewer 2’s comment, we have made the following changes to the text of the manuscript on page 7, “Results”, first paragraph of “Study participants”, 3rd and 4th sentence, to include median, IQR, min and max for age and temperature:

“The median age (IQR, min, max) of participants was 58 (49 – 66, 40, 90) years, with a 2:1 ratio of males to females. The median axillary body temperature (IQR, min, max) at study entry was 38.2°C (38.0°C – 38.4°C, 37.8°C, 40.8°C) in the SoC+Doxy group and 38.3° C (37.6°C – 38.3°C, 37.6°C, 40.6°C) in the SoC group.”

In the revised Table 1, we summarise the following variables:

Variable Comment

Age Continuous, but not Normal. We currently report mean and range. I’ve added the requested summaries in reviewer2-requests.pdf

Sex Dichotomous. We currently report n and percentage. I suggest no change

Baseline PCR status Dichotomous. We currently report n and percentage. I suggest no change

Body temp Continuous. Is it Normal? Probably not. I’ve added the requested summaries in reviewer2-requests.pdf

Comorbidities Dichotomous. We currently report n and percentage. I suggest no change

Con meds Dichotomous. We currently report n and percentage. I suggest no change

Table 2: Suggest no changes: all variables are dichotomous.

Final note to academic editor

Thank You for all your efforts with our manuscript and for considering this revised version for publication in PLoS One.

---

## [Editor Report · Decision Letter 2]

8 Jan 2023

Doxycycline for the prevention of progression of COVID-19 to severe disease requiring intensive care unit (ICU) admission: a randomized, controlled, open-label, parallel group trial (DOXPREVENT.ICU)

PONE-D-22-10587R2

Dear Dr. Gadola,

We’re pleased to inform you that your manuscript has been judged scientifically suitable for publication and will be formally accepted for publication once it meets all outstanding technical requirements.

Kind regards,

Davor Plavec, MD, MSc, PhD, Prof.

Academic Editor

PLOS ONE

Additional Editor Comments (optional):

The manuscript is acceptable in its current form.
---

## [Editor Report · Acceptance letter]

13 Jan 2023

PONE-D-22-10587R2 

Doxycycline for the prevention of progression of COVID-19 to severe disease requiring intensive care unit (ICU) admission: a randomized, controlled, open-label, parallel group trial (DOXPREVENT.ICU) 

Dear Dr. Gadola:

I'm pleased to inform you that your manuscript has been deemed suitable for publication in PLOS ONE. Congratulations! Your manuscript is now with our production department. 

Kind regards, 

on behalf of

Dr. Davor Plavec 

Academic Editor

PLOS ONE